# Deep Stacking Network for Intrusion Detection

**DOI:** 10.3390/s22010025

**Published:** 2021-12-22

**Authors:** Yifan Tang, Lize Gu, Leiting Wang

**Affiliations:** School of Cyberspace Security, Beijing University of Posts and Telecommunications, Beijing 100876, China; tyfcs@bupt.edu.cn (Y.T.); wlt562502@bupt.edu.cn (L.W.)

**Keywords:** intrusion detection, ensemble learning, decision tree, deep neural network, deep stacking network, nsl-kdd

## Abstract

Preventing network intrusion is the essential requirement of network security. In recent years, people have conducted a lot of research on network intrusion detection systems. However, with the increasing number of advanced threat attacks, traditional intrusion detection mechanisms have defects and it is still indispensable to design a powerful intrusion detection system. This paper researches the NSL-KDD data set and analyzes the latest developments and existing problems in the field of intrusion detection technology. For unbalanced distribution and feature redundancy of the data set used for training, some training samples are under-sampling and feature selection processing. To improve the detection effect, a Deep Stacking Network model is proposed, which combines the classification results of multiple basic classifiers to improve the classification accuracy. In the experiment, we screened and compared the performance of various mainstream classifiers and found that the four models of the decision tree, k-nearest neighbors, deep neural network and random forests have outstanding detection performance and meet the needs of different classification effects. Among them, the classification accuracy of the decision tree reaches 86.1%. The classification effect of the Deeping Stacking Network, a fusion model composed of four classifiers, has been further improved and the accuracy reaches 86.8%. Compared with the intrusion detection system of other research papers, the proposed model effectively improves the detection performance and has made significant improvements in network intrusion detection.

## 1. Introduction

With the development of the Internet of Things (IoT), device embedding and connection have generated more and more network data traffic [1]. The increase in data volume has also led to more threats to network security. With the updating of network technology, more and more malicious attacks and threat viruses are appearing and spreading at a faster speed [2]. As the main means to defend against advanced threats, network intrusion detection faces new challenges. There are two common detection methods: feature-based detection and anomaly-based detection [3]. When the attack signature is known, signature-based detection is very useful. Conversely, anomaly-based detection can be used for known or unknown attacks. As a traditional network attack detection method, the intrusion detection system based on feature detection is widely used because of its simplicity and convenience. Its shortcomings are also obvious. The feature-based intrusion detection system cannot detect unknown attack types and the detection accuracy is limited by the feature size and update speed of the signature database. In recent years, researchers have tried to introduce other technologies in intrusion detection to solve this problem, especially the recent emergence of machine learning technology. Many researchers have applied machine learning algorithms, such as decision trees, k-nearest neighbors, support vector machines and deep neural networks to the field of intrusion detection and have achieved some initial results.

However, according to the ‘no free lunch’ theorem, we cannot find the best algorithm [4]. Each algorithm model may be outstanding in some aspects and inferior to other algorithms in some parts. Most traditional machine learning methods often have many problems, such as low versatility, slow detection time, insufficient detection accuracy and so on. So we try to use integrated learning to solve this problem. In addition, many studies only focus on the overall detection accuracy and the detection effect for a few types of data is often very low. How to strengthen the detection ability of small-scale samples is also one of the problems that need to be solved at present.

This paper proposes a deep stacking network model. This model belongs to an ensemble learning model, which integrates the advantages of various machine learning algorithms and improves the detection rate of various attack categories in network intrusion detection. The main contributions of this paper are summarized as follows:An ensemble learning system DSN is proposed, consisting of the decision tree, k-nearest neighbor, random forest and deep neural network. DSN improves the accuracy of intrusion detection technology and provides a new research direction for intrusion detection.The proposed DSN combines the predictions of multiple basic classifier models, fusing decision information and improving the generalization and robustness of the detection model.We use a real NSL-KDD dataset to evaluate our proposed system. The experimental results show that DSN has better performance than traditional methods and most current algorithms. We consider that the proposed system has good application prospects for IDS.

The rest of the paper organizes as follows: Section 2 briefly reviews intrusion detection technology. Section 3 describes the dataset and the algorithms used, including decision tree, deep neural network and deep stacking network. In Section 4, the proposed DSN algorithm is described in detail. The experiments of choosing basic classifiers and comparison experiments of performance analysis are given respectively in Section 5. Finally, Section 6 provides some personal opinions and conclusions, including further work afterward.

## 2. Related Works

In recent years, many scholars have tried to use machine learning algorithms to study new intrusion detection methods [5]. Several studies have suggested that by selecting relevant features, the detection accuracy and performance of IDS can be considerably improved [6]. Hodo et al. [7] analyzed the advantages of various machine learning methods in intrusion detection and discussed the influence of FS in IDS. Janarthanan et al. [8] conducted experiments to compare the effects of features on various machine learning algorithms and pointed out some most important features in intrusion detection. Some scholars focus on using Feature selection (FS) to improve intrusion detection performance. Bamakan et al. [9] presented a novel support vector machine (SVM) with FS by Particle swarm optimization (PSO), which improved the performance of classification for IDS. Elmasry et al. [10] applied two PSO algorithms to perform feature selection and hyperparameter selection respectively, which improved the detection effect of deep learning architectures on IDS. Thaseen et al. [11] designed a multiclass SVM with chi-square feature selection, which reduces the time of training considerably and effectively improves the efficiency of the algorithm.

Deep learning has achieved many successes in speech detection, image recognition, data analysis and other fields, becoming the preferred solution to many problems. Many scholars have also begun to use deep learning to solve intrusion detection problems. Wu et al. [12] designed a convolutional neural network (CNN) to select features from data sets automatically and set the weight coefficient of each class to solve the problem of sample imbalance. Muhammad et al. [13] proposed the IDS based on a stacked autoencoder (AE) and a deep neural network (DNN), which reduced the difficulty of network training and improved the performance of the network. Yang et al. [14] designed a DNN with an improved conditional variational autoencoder (ICVAE) to extract high-level features, overcome some limitations of shallow learning and further promote the progress of intrusion detection systems.

Although machine learning and deep learning have certain advantages in intrusion detection, the disadvantages are also obvious. A single algorithm tends to have a high detection rate for certain attack categories while ignoring the detection effect of other attack categories. In order to solve this problem, many scholars try to use the idea of integrated learning to solve the problem of intrusion detection. Rahman et al. [15] proposed an adaptive intrusion detection system based on boosting with naive Bayes as the weak (base) classifier. Syarif [16] applied and analyzed three traditional ensemble learning methods for intrusion detection. Gao [17] proposed an adaptive voting model for intrusion detection, which consists of four different machine learning methods as the base classifiers, resulting in an excellent performance.

The development of the above-mentioned intrusion detection technologies is encouraging, but these classification technologies still have detection deficiencies, such as being insensitive to unknown attacks and a low detection rate when detecting a few attacks. In order to overcome these problems, this paper uses preprocessing technology to deal with the dataset and selects the basic classifier of ensemble learning selected to construct the ensemble learning model DSN. Finally, the system DSN solves the above-mentioned problems by learning the advantages of each classifier.

## 3. Background

### 3.1. NSL-KDD Dataset Introduction

The famous public KDDCUP’ 99 is the most widely used data set for the intrusion detection system [18]. However, there are two critical problems with this data set, which seriously affect the performance of the evaluated system. One is that many redundant duplicate records will cause the learning algorithm to be biased towards identifying duplicate records. Second, the sample ratio is seriously unbalanced and some attack categories exceed 70%, making them too easy to be detected, which is not helpful for multi-class detection. Both of these problems have seriously affected the evaluation of intrusion detection performance. To solve these problems, Tavallaee proposed a new data set NSL-KDD [19,20], which consists of selected records of the complete KDD data without mentioned shortcomings [5]. Table 1 shows the detailed information of the dataset NSL-KDD.

Many scholars have carried out a series of studies on NSL-KDD and analysis shows that the NSL-KDD data set is suitable for evaluating different intrusion detection models [21]. Therefore, we selected the NSL-KDD data sets to validate the proposed model. Table 1 shows the distribution of the NSL-KDD dataset.

### 3.2. Decision Tree

Decision tree (DT) is a commonly used machine learning method to complete classification and regression tasks. The decision tree model has a tree structure, starting from the root node and branching using the essential features of the data. Each branch represents the output of a feature and each child node represents a category. The classification decision tree is a kind of supervised learning and the required classification model can be obtained by giving sample training. The input data finally completes the classification task through the judgment of each node. According to the criteria for judging branch characteristics, decision trees can be divided into ID3, C4.5 and CART. ID3 uses a greedy strategy and uses information gain based on information entropy as a branch criterion.

In the classification problem, take a data set D with K classes as an example. The information entropy of probability distribution is defined as follows:(1)$$Ent\left(D\right)=\overset{K}{\underset{k=1}{\sum }}p_{k}\log_{2}\left(p_{k}\right)$$
where $p_{k}$ represents the probability of sample points belonging to k class.

Choose feature A as the split node, the conditional entropy and information gain is defined as follows:(2)$$Ent\left(D|A\right)=\overset{J}{\underset{j=1}{\sum }}\frac{\left|D_{j}\right|}{\left|D\right|}Ent\left(D_{j}\right)$$
(3)Gain(D,A)=Ent(D)−Ent(D|A)
where $D_{j}$ represents the sample subset of class j in feature A.

The greater the information entropy, the higher the uncertainty of the sample set. The essence of the classification learning process is the reduction of sample uncertainty (that is, the process of entropy reduction). The greater the change of the information gain, the better the classification effect of the feature on the sample set. Therefore, the feature split with the largest information gain should be selected.

### 3.3. Deep Neural Network Algorithm

Deep neural network (DNN) is a deep learning algorithm widely recognized by scholars. Figure 1 shows the basic structure of DNN. The network structure of DNN includes the input layer, hidden layer and output layer and each layer is fully connected. Each neuron has no connection with the neurons between the layers and is connected with all the neurons in the next layer. After each layer of the network, there is an activation function acting on the output, which strengthens the effect of network learning. Therefore, DNN can also be understood as a large perceptron composed of multiple perceptrons. Take the ith layer forward propagation calculation as an example, the formula is as follows:(4)$$x_{i+1}=\sigma \left(\sum^{\;}w_{i}x_{i}+b\right)$$
where x represents the input value, w represents the weight coefficient matrices and b represents the bias vector.

In a multi-class network, ReLU is usually used as an activation function, the formula is as follows:(5)σ(x)=max(0,x)

The loss function measures the output loss of training samples and calculates the back propagation of the network through the loss function to optimize the network structure. In the classification task, we usually choose cross-entropy as the loss function, the formula is as follows:(6)$$C=-\frac{1}{N}\sum_{x}\sum_{i=1}^{M}\left(y_{i}\log p_{i}\right)$$
where N represents the number of the input data set, M represents the number of categories, $y_{i}$ represents whether the classification i corresponds to the real category and $p_{i}$ represents the probability of predicting into category i.

### 3.4. Deep Stacking Network Algorithm

Individual machine learning algorithms usually have shortcomings and cannot complete complex task requirements. Therefore, we try to combine many different machine learning algorithms to form a learning system. We call this learning system ensemble learning and the algorithms that make up the learning system are called individual learners. Ensemble learning can be divided into two categories. One type is serialization methods that have strong dependencies between individual learners and must be generated serially, such as boosting and AdaBoost. The other is parallelization methods that can be generated at the same time without strong dependencies on individual learners, such as bagging and random forest (RF).

Stacking is a combination strategy that combines the calculation results of individual learners. Wolpert [22] put forward the idea of stacked generalization in 1992, using another machine learning algorithm to combine the results of individual machine learning devices. This method improves the performance of the algorithm, reduces the generalization error and makes the model more widely used. Deng proposed the use of deep neural networks as the combined layer algorithm to further improve the performance of the stacking model, called the deep stacking network (DSN) [23].

DSN usually consists of two modules. The first module is the classifier module, composed of classifiers with different classification performances and performs preliminary prediction processing on the input data. ‘Stacking’ refers to concatenating all output predictions with the original input vector to form a new input vector for the next module. The second module is the prediction fusion module. By training the new combined input data obtained from the previous layer, a new network can be obtained. The network can effectively use the output data obtained from the previous layer for further processing. The prediction result output by the network is more accurate and closer to the true value.

## 4. The Proposed Intrusion Detection Method

In this paper, a deep stacking network model is designed which selects commonly used machine learning algorithms, such as support vector machines (SVM), decision trees, random forests, k-nearest neighbors (KNN), AdaBoost, deep neural networks (DNN), etc., as the basic classifiers. Through comparative testing, we select four machine learning methods as the basic classifiers. Through data preprocessing and deep neural network tuning, the best detection effect is finally obtained. Figure 2 shows the algorithm flow of the proposed model, mainly includes following 7 steps:Input the original NSL-KDD training data set. The pre-processing module discretizes the string information in the data set, filters important feature selection, handles imbalanced data and normalizes the data.Use 10-fold cross-validation to divide the pre-processed dataset and then input the data into various classifiers for training.After using training data to conduct cross-validation training for all algorithms, select algorithms with better detection accuracy and operational performance as the basic classifier. Then discretize the classification results.Input the predicted classification result of the training set and the original category as the training set, initialize the parameter weights of the neural network, train the network parameters and finally generate the Deep Stacking Network model.Input the NSL-KDD testing data set. The pre-processing module discretizes the character information in the data set, selects essential features and normalizes the data.Use the trained basic classifier to initially predict the classification results and discretize the results.Input the preliminary predicted classification results into the trained neural network to obtain the prediction results of the Deep Stacking Network model.

### 4.1. Data Pre-Processing

Data pre-processing is a necessary step for data analysis, and it is also an essential part of an intrusion detection system. The preprocessing stage mainly includes four units: one-hot encoding, feature selection, data standardization, imbalance handling and normalization.

#### 4.1.1. One-Hot-Encoding

There are 41 features in the NSL-KDD dataset, including 3 string features and 38 continuous value features. In machine learning, character type information cannot be used directly and needs the encoding methods to convert it. One-hot-encoding is one of the most commonly used methods to deal with the numeralization of categorical features [24]. It converts each character type feature into a binary vector and marks the corresponding category as 1 and the others as 0. For example, the feature *protocol_type* has a total of three attributes: *tcp*, *udp* and *icmp*. By one-hot-encoding, *tcp* is encoded into (1, 0, 0), *udp* is encoded into (0, 1, 0) and *icmp* is encoded into (0, 0, 1). Overall, the three character type features *protocol_type*, *service*, *flag* are mapped into 84-dimensional binary values. Otherwise, the *num_outbound_cmds* feature value is 0, so this feature is removed. Therefore, the original 41-dimensional NSL-KDD can be transformed into a new 121-dimensional data set.

#### 4.1.2. Feature Selection

Feature selection (FS) is a commonly used method of data aggregation. In some ways, we can select important features and remove the remaining redundant features to alleviate the problem of dimensionality disaster. Similarly, removing irrelevant features can reduce the difficulty of machine learning tasks and increase the efficiency of storage space utilization. In some machine learning algorithms, FS can help the algorithm improve detection performance, especially decision tree algorithms [25].

#### 4.1.3. Imbalance Handling

It can be clearly seen from the table that the training samples are imbalanced on the NSL-KDD data set. Unbalanced training samples will cause the trained model to be biased to recognize most sample categories, resulting in the degradation of the model’s detection performance. Therefore, we choose to process the training samples. Commonly used methods for processing unbalanced data sets include under-sampling and over-sampling. The model in this paper uses undersampling to process the training samples of the data set. From a security perspective, the intrusion detection system should identify the attack type as much as possible and can appropriately reduce the normal traffic data in the training sample when inputting the training sample, so that the focus of model training is to identify the attack type. We use the non-replacement method to sample the normal flow data randomly. Table 2 shows the sample distribution of the new data set.

#### 4.1.4. Normalization

Different dimensions of input data usually have different dimensions and orders of magnitude. When using machine learning, data normalization is a very necessary measure. The transformed NSL-KDD has 121-dimensional features and there are big differences between the features, so we use data normalization to reduce the differences for improved performance [26]. In this paper, the zero-mean normalization and the min–max normalization method are adopted to reduce the differences in different dimensions. The zero-mean normalization processes the data by changing the average value to 0 and the standard deviation to 1. The formula is as follows:(7)$${\tilde{Z}}_{ij}=\frac{Z_{ij}-\bar{Z_{i}}}{\sigma }\;$$
where $\bar{Z_{i}}$ and σ, respectively, represent the mean and standard deviation value of the ith feature $Z_{i}$ and ${\tilde{Z}}_{ij}$ represents the feature value after normalization.

The min-max normalization scales the data to the interval [0, 1] through a linear transformation. The formula is as follows:(8)$${\tilde{Z}}_{ij}=\frac{Z_{ij}-\min\left(Z_{i}\right)}{\max\left(Z_{i}\right)-\min\left(Z_{i}\right)}$$
where $\max\left(Z_{i}\right)$ and $\min\left(Z_{i}\right)$, respectively, represent the maximum and minimum value of the ith feature $Z_{i}$ and ${\tilde{Z}}_{ij}$ represents the normalized feature value between [0, 1].

### 4.2. Training Classifiers

The classifier module reads the preprocessed data and uses the ten-fold cross-validation method to process the training data. In the 10-fold cross-validation method, the entire training set is randomly divided into 10 folds, of which 9 folds work as sub-training data and the remaining 1 fold works as sub-validation data. The read data is used for model training. We first choose KNN, RF, SVM, DT, LR, DNN to process the data. Two standardization methods are mentioned in Section 4.1. According to the different characteristics of machine learning algorithms, we have different data standardization methods for different algorithms. Most studies have proved that feature selection can improve the effectiveness of decision tree algorithms [26]. Therefore, in the setting of the decision tree algorithm, the feature selection method is used for feature selection. Commonly used feature selection methods include the correlation coefficient method, PSO feature selection method, etc. In this paper, the PSO method is selected as the feature selection method [27]. Table 3 shows the different preprocessing methods of different algorithms:

The classification decision tree in this article uses the ID3 algorithm as the way to build the tree model. First, perform feature selection, reducing the number of features of the input data from 121 to 56. Then use the data after feature selection to input decision tree training. Each time the feature with the largest information gain is selected as the bifurcation node, each child node connects two branches to build a binary decision tree. In the decision tree, choosing the bifurcation point is the key to affecting the classification performance of the decision tree. Take the 4-layer decision tree trained on NSL-KDD data as an example, where *src_bytes* is the root node of the decision tree and the decision tree model is shown in Figure 3.

The settings of different DNN network structures usually affect the results of training, so this paper designs a DNN network structure with five hidden layers. According to the proportion of the number of samples in the training sample, we reset the proportion of each category class_weight = {1:2:3:4:5} to increase the effect of small samples in network training, thereby improving the overall detection of the DNN network Effect. Finally, use the softmax function to output the final category prediction results. Figure 4 shows the network structure settings of DNN.

### 4.3. Proposed Deep Stacking Network

The Deep Stacking Network (DSN) is divided into two layers. In the first layer, the classifier module, each classifier has 10 different model parameter structures based on 10-fold cross-validation. Each model performs a result prediction on its corresponding verification set and can get the prediction results of 10 verification sets. The set of 10 validation sets corresponds to a complete training set. We superimposed the prediction results of the 10 validation sets to obtain the prediction results of a complete training set. This prediction result on the training set can help us evaluate the performance of the classifier and use it as the new training set input to the next layer. At the same time, each model inputs the data that need to be predicted and the mode of the predicted values of these 10 models are taken as the prediction result of the classifier, which is used as the new test set to input by the next layer. Figure 5 shows the process of data ‘stacking’. So far, we have not only made full use of the training effect of the complete training set but also used the entire training set for model performance evaluation.

In the algorithm proposed in this paper, 4 classifiers are selected as the basic classifiers of Deep Stacking Network. Each classifier corresponds to a new training set and a new test set. Then use one-hot encoding to convert the prediction results from character variables to discrete variables and the prediction results from 1-dimensional features to 5-dimensional features. For example, the prediction result is that Probe becomes (0, 0, 1, 0, 0). Therefore, a total of 20-dimensional training set features and 20-dimensional test set features can be obtained. Combining the original classification label of the training set with the features of the new training set is the input of the new training set. The features in this training set can also be called appearance features. These appearance features can help us train to get the influence relationship of each basic classifier in the network. Figure 6 shows the composition of the new training set and test set.

In the second layer, the prediction fusion module, we use a simple neural network for decision fusion. Since decision fusion does not need to explore the deep relationship between features and labels, we use a hidden layer of neural network for model training. The neural network structure is shown in Figure 7. Adjust the network parameters by inputting a new training set, use the ReLU function as the activation function and use the softmax function to adjust before outputting to get the final classification result.

## 5. Experimental Results and Analysis

### 5.1. Performance Evaluation

In this article, nine indicators commonly are used in intrusion detection to evaluate the performance of the intrusion detection system, including four confusion matrix indicators of true positive (TP), true negative (TN), false positive (FP), false negative (FN) and five evaluation indicators of accuracy (ACC), precision, recall rate, F1-score, multi-class accuracy (MACC). Table 4 shows the confusion matrix.

The four confusion matrix indicators are defined as follows:True Positive (TP): Attack records that are correctly detected as attack ones.False Positive (FP): Normal records that are incorrectly detected as attack ones.True Negative (TN): Normal records that are correctly detected as normal ones.False Negative (FN): Attack records that are incorrectly detected as normal ones.

The six evaluation indicators are defined as follows:(9)$$\mathrm{Accuracy}=\frac{\mathrm{TP}+\mathrm{TN}}{\mathrm{TP}+\mathrm{TN}+\mathrm{FP}+\mathrm{FN}}$$
(10)$$\mathrm{Precision}=\frac{\mathrm{TP}}{\mathrm{TP}+\mathrm{FP}}$$
(11)$$\mathrm{Recall}=\frac{\mathrm{TP}}{\mathrm{TP}+\mathrm{FN}}$$
(12)$$F1-\mathrm{score}=\frac{2\mathrm{TP}}{2\mathrm{TP}+\mathrm{FN}+\mathrm{FP}}$$

The ACC is usually an indicator of traditional binary classification tasks. According to the standard of multi-attack classification, multi-class accuracy (MACC) is proposed, which can help us better compare the performance of classifiers.
(13)$$\mathrm{MACC}=\frac{\mathrm{Number}\;\mathrm{of}\;\mathrm{samples}\;\mathrm{successfully}\;\mathrm{classified}}{\mathrm{Total}\;\mathrm{number}\;\mathrm{of}\;\mathrm{samples}}$$

### 5.2. Experimental Setup

The proposed system is performed by a laboratory computer with Intel(R) Core(TM) i7-9750H CPU@ 2.60 GHz and 16.00 GB of RAM using Python on system Windows 10. All experiments are performed on the preprocessed NSL-KDD dataset. Firstly, select the appropriate basic classifier by screening the appropriate machine learning algorithm. After selecting the basic classifier, we conduct experiments on the complete system to evaluate the performance of the model.

Figure 3 and Figure 6 show the structures of two neural networks used in the system, DNN and DSN. The number of neurons in the hidden layer in DNN is 2048-1024-512-256-128, the number of neurons in the hidden layer in DSN is 128, the activation function of the hidden layer is ReLU and the activation function of the output layer is Softmax. The optimization algorithm of two networks is Adam [28], where two important parameters need to be set, named the learning rate and the number of epochs.

When the learning rate of the network is too high, the loss function of networks will oscillate without convergence. If the learning rate is too low, the slow convergence rate will hinder the update of networks. Therefore, choosing an appropriate learning rate is very important for network performance optimization. In this experiment, a set of learning rates [0.1, 0.01, 0.001, 0.0001, 0.00001] is selected as the candidate parameters of the two networks and the accuracy of the network on the verification set is used as the measurement standard. Similarly, the number of iterations is also critical to the optimization of the network. A large number of epochs will cause the network to waste time cost and it is easy to cause the network to overfit. The small number of epochs will result in insufficient network convergence and poor model learning performance. This experiment finds the appropriate number of iterations from the changing law of the loss function value during network training.

In order to find the right parameters, we use the 10-fold cross-validation method mentioned in Section 4.2 to find the best parameters. For the basic classifier DNN, as shown in Figure 8, the learning rate is optimal between 0.0001 and 0.00001 and finally set to 0.00003. As shown in Figure 9, the experiment shows that the training loss basically does not change after 50 iterations. We set the number of iterations to 50. For DSN, as shown in Figure 10, the learning rate reaches the maximum accuracy at 0.001. We choose 0.001 as the learning rate. As shown in Figure 11, the loss function of the network stabilizes after 20 iterations, so we choose to set the number of iterations to 20.

In the feature selection of DT, try to select a different number of features to test the classification effect. As shown in the Figure 12, when the number of features is 56, the best accuracy of 99.78% can be achieved. Therefore, the number of DT feature selections in this article is set to 56. The parameters of other basic classifiers are set according to the default parameters provided by the Sklearn library.

In order to establish a good ensemble learning model, it is first necessary to screen the basic classifiers with excellent performance. In the experiment, a 10-fold cross-validation method was used to evaluate the performance of the six selected algorithms. We consider the effect of the algorithm from the perspective of the predicted success rate of each attack type so that the characteristics of each classifier can be analyzed, which will help us choose a good basic classifier to improve the performance of the entire intrusion detection system. Table 5 shows the results of cross-validate on the new training set.

From the table, it is appreciated that three algorithms of KNN, DT and RF have outstanding performance in detection accuracy. Among them, RF has the best performance in detecting Normal categories, DT has the best performance in detecting Probe and R2L categories and DNN has the best performance in detecting DoS and U2R categories. In terms of time spent, DT used the shortest time and SVM used the longest due to slow modeling. Stacked generalization requires us to choose the classifiers to be good and different, so we choose KNN, DT, RF and DNN that have outstanding performance in various aspects as the basic classifier of the DSN network.

### 5.3. Results and Discussion

Table 6 and Table 7 respectively show the performance of each classifier on the test set and the performance results of the DSN model on the NSL-KDD test set. From the perspective of accuracy, DT and DNN have reached high accuracy. Among them, RF has the best performance in detecting Normal categories, DT has the best performance in detecting Probe and R2L categories and DNN has the best performance in detecting DoS and U2R categories. This is basically the same as the previous results on the validation set and meets the different requirements of a good classifier. The DSN model is not prominent in each attack category, but it combines the advantages of four basic classifiers, improves the overall classification accuracy and also solves the problem of the low accuracy of a single algorithm in certain categories of attack recognition. In terms of training and testing time, the proposed model is acceptably higher than most algorithms and lower than SVM. The multi-class detection accuracy of DSN reached 86.8%, the best performance. 

In order to better demonstrate the performance of this system in intrusion detection, we will compare the proposed model with the intrusion detection algorithms proposed by seven scholars, including DNN, RNN, Ensemble Voting and SAAE-DNN. Table 8 shows the classification accuracy of the algorithm on NSL-KDD Test+ and NSL-KDD Test-21, respectively. The classification accuracy of DSN on NSL-KDD Test+ is 86.8% and the classification accuracy on NSL-KDD Test-21 is 79.2%, which is significantly higher than other comparison algorithms.

## 6. Conclusions and Future Work

This paper proposes a novel intrusion detection approach called DSN that integrates the advantage of four machine learning methods. For the real network dataset NSL-KDD, we use Pre-processing to normalize data. In the experiment, four of the six machine learning methods were selected as the basic classifiers for ensemble learning. The integrated learning model DSN gathers the advantages of four different classifiers and improves the performance of the algorithm. Compared with other researches, it is proved that our ensemble model effectively improves the detection accuracy. The DSN proposed in this article has a good application prospect, which is worthy of further exploration.

The data used in the experiment is NSL-KDD, which is an unbalanced data set. Therefore, the use of this data set for training will inevitably lead to the learning result biased towards the majority of samples. How to use limited training samples to improve the adaptability of multi-classification is the key to solving the problem. Ensemble learning is an excellent method that can improve the performance of the model in a short time. However, it is not advisable to use ensemble learning methods blindly. Different algorithms are suitable for different classification situations. It is necessary to select the correct algorithm as the basic classifier to fundamentally improve the overall effect of the model. From the condition of model optimization, the most important thing is to optimize the data, followed by the optimization algorithm and finally, the parameters of the optimization algorithm.

Future work will focus on improving IDS performance. I consider the detection cost of each algorithm for different attack categories as a measurement standard and design a new IDS. At the same time, I will choose other intrusion detection data sets collected from reality to experiment with the intrusion detection performance of the algorithm. Designing an intrusion detection system capable of parallel processing and learning is our next step of work.

## Figures and Tables

**Figure 1 sensors-22-00025-f001:**
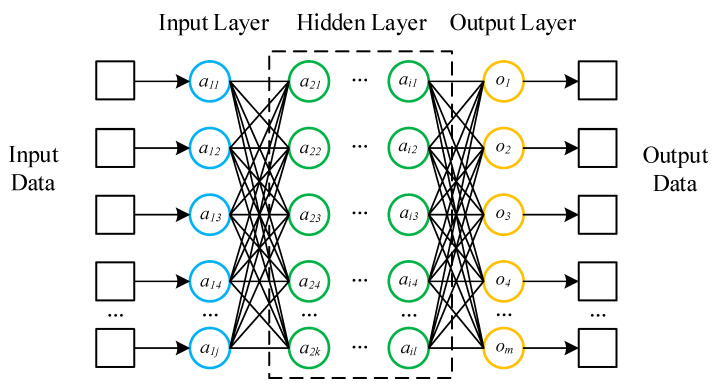
The basic structure of DNN.

**Figure 2 sensors-22-00025-f002:**
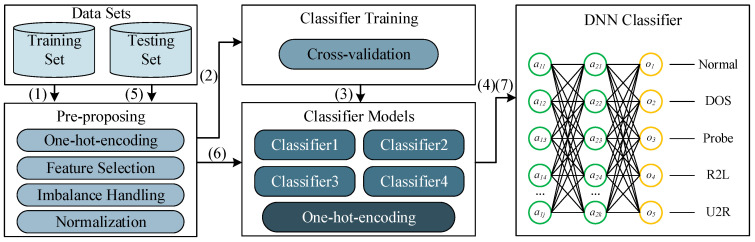
Deep stacking network model.

**Figure 3 sensors-22-00025-f003:**
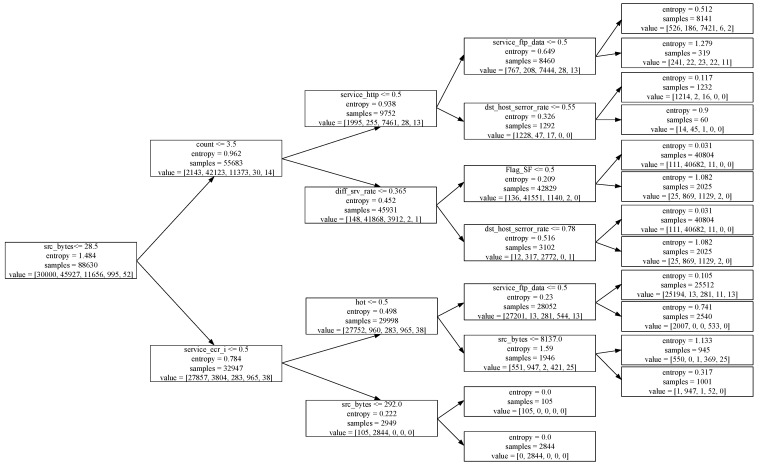
4-level binary decision tree model.

**Figure 4 sensors-22-00025-f004:**
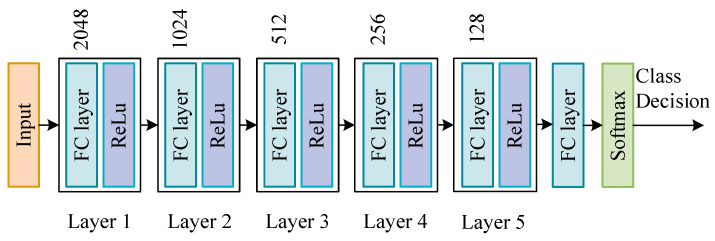
DNN model structure settings.

**Figure 5 sensors-22-00025-f005:**
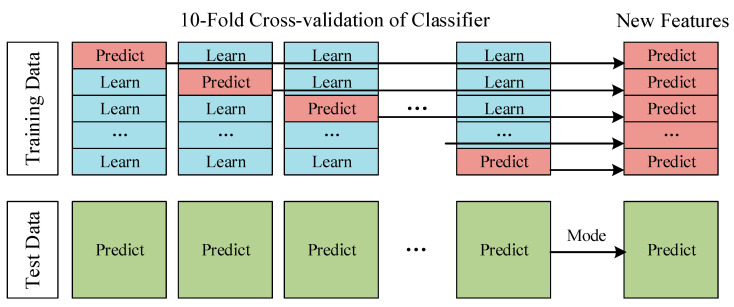
10-fold cross-validation.

**Figure 6 sensors-22-00025-f006:**
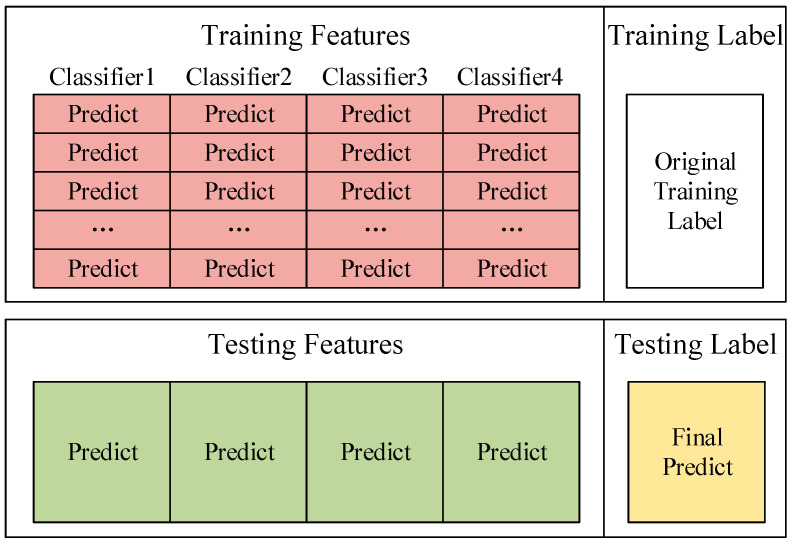
The composition of the new training set and test set.

**Figure 7 sensors-22-00025-f007:**
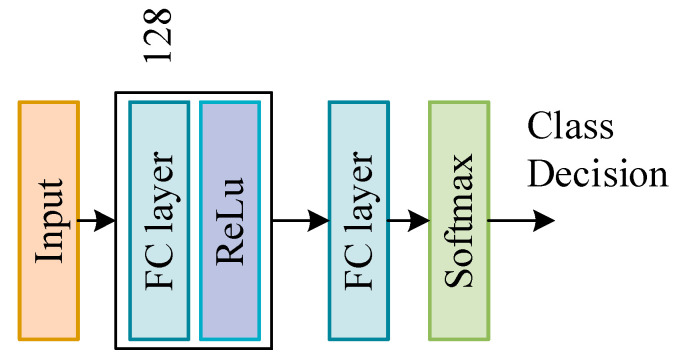
DSN model structure settings.

**Figure 8 sensors-22-00025-f008:**
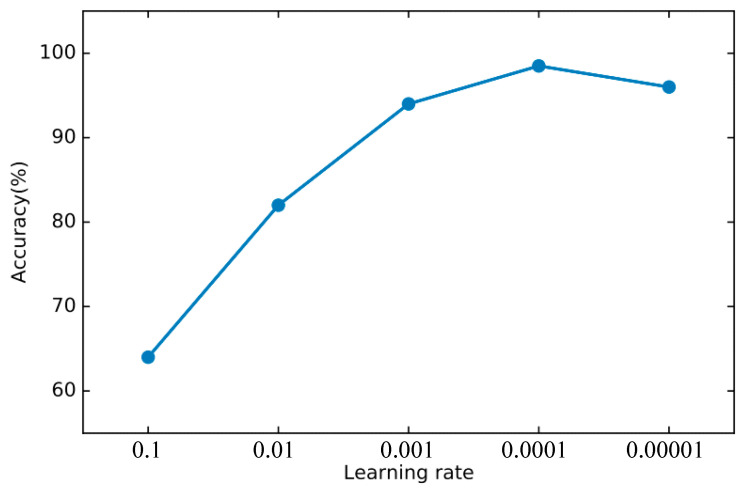
The accuracy of DNN changes when different learning rates are set.

**Figure 9 sensors-22-00025-f009:**
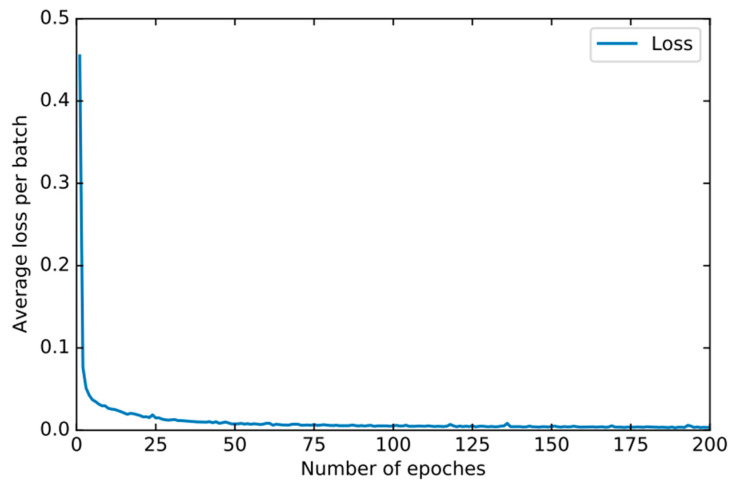
The training loss of DNN changes when different epochs are set.

**Figure 10 sensors-22-00025-f010:**
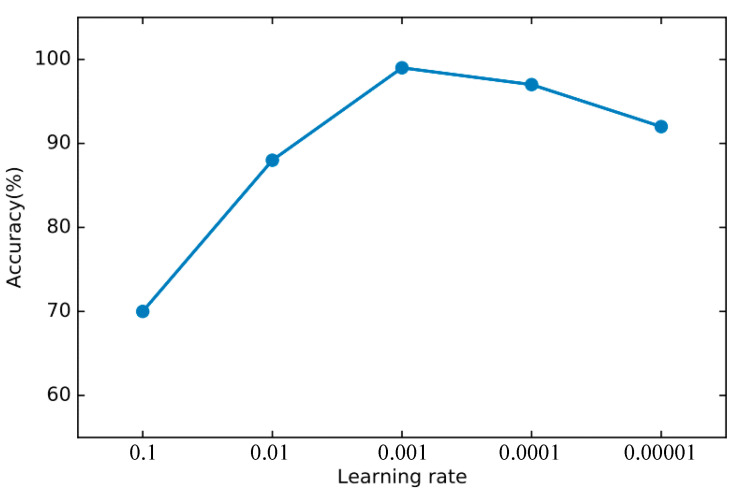
The accuracy of DSN changes when different learning rates are set.

**Figure 11 sensors-22-00025-f011:**
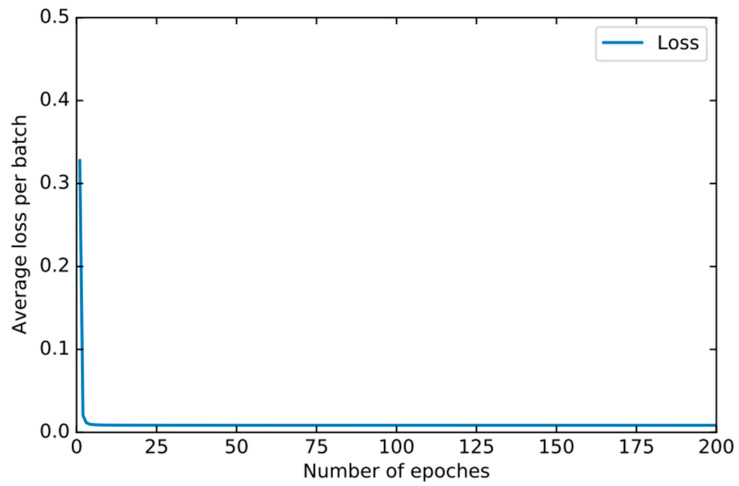
The training loss of DSN changes when different epochs are set.

**Figure 12 sensors-22-00025-f012:**
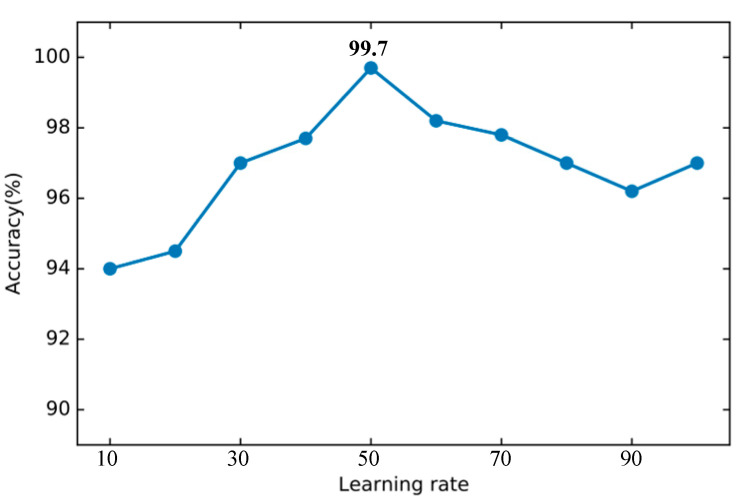
Accuracy of decision tree using different number of features.

**Table 1 sensors-22-00025-t001:** The class distribution of the NSL-KDD dataset.

Category	Attack	Training Dataset	Testing Dataset	
		KDD Train+	KDD Test+	KDD Test-21
normal	normal	67,343	9711	2152
Subtotal		67,343	9711	2152
DoS	neptune	41,214	4657	1579
	smurf	2646	665	627
	back	956	359	359
	teardrop	892	12	12
	pod	201	41	41
	land	18	7	7
	apache2	/	737	737
	processtable	/	685	85
	mailbomb	/	293	293
	udpstorm	/	2	2
Subtotal		45,927	7458	4342
Probe	satan	3633	735	727
	ipsweep	3599	141	141
	portsweep	2931	157	156
	nmap	1493	73	73
	mscan	/	996	996
	saint	/	319	309
Subtotal		11,656	2421	2402
R2L	warezclient	890	/	/
	guess_passwd	53	1231	1231
	warezmaster	20	944	944
	imap	11	1	1
	ftp_write	8	3	3
	multihop	7	18	18
	phf	4	2	2
	spy	2	/	/
	named	/	17	17
	sendmail	/	14	14
	xlock	/	9	9
	xsnoop	/	4	4
	worm	/	2	2
	snmpgetattack	/	178	178
	snmpguess	/	331	331
Subtotal		995	2754	2754
U2R	buffer_overflow	30	20	20
	rootkit	10	13	13
	loadmodule	9	2	2
	perl	3	2	2
	httptunnel	/	133	133
	ps	/	15	15
	xterm	/	13	13
	sqlattack	/	2	2
Subtotal		52	200	200
Total		125,972	22,544	11,850

**Table 2 sensors-22-00025-t002:** Number of New Records.

Category	Number of Original Records	Number of New Records
Normal	67,343	30,000
DoS	45,927	45,927
Probe	11,656	11,656
R2L	995	995
U2R	52	52
Total	125,972	88,630

**Table 3 sensors-22-00025-t003:** Pre-processing methods of different algorithms.

Algorithm	KNN	DT	RF	SVM	LR	DNN
Normalization Method	min-max	/	/	min-max	min-max	zero-mean
Feature Selection	No	Yes	No	No	No	No
One-Hot-Encoding	Yes	Yes	Yes	Yes	Yes	Yes
Imbalance Handling	Yes	Yes	Yes	Yes	Yes	Yes

**Table 4 sensors-22-00025-t004:** Confusion Matrix.

	Predicted Attack	Predicted Normal
Actual Attack	TP	FN
Actual Normal	FP	TN

**Table 5 sensors-22-00025-t005:** Detection performance of cross-validate on new training set.

Model	Normal	DoS	Probe	R2L	U2R	MACC	Time (s)
KNN	99.78%	99.91%	99.51%	94.17%	59.62%	99.72%	82
DT	99.7%	99.97%	99.65%	96.68%	67.31%	99.78%	0.7
RF	99.79%	99.93%	99.62%	95.68%	42.31%	99.76%	1.2
SVM	99.5%	99.88%	98.92%	94.47%	34.61%	99.53%	1212
DNN	98.75%	99.99%	99.55%	95.97%	73.07%	99.57%	162
LR	98.92%	99.25%	98.05%	87.23%	38.46%	98.8%	33

**Table 6 sensors-22-00025-t006:** Detection performance for different classifiers based on the NSL-KDD testing+.

Model	Normal	DoS	Probe	R2L	U2R	ACC	MACC	Recall	Precision	F1-Score	Time (s)
KNN	92.78%	84.88%	71.29%	5.08%	9%	78.81%	76.41%	64.02%	92.18%	75.56%	283
DT	96.81%	85.43%	91.6%	50.5%	13%	89.56%	86.1%	78.01%	97.37%	86.62%	1.7
RF	97.41%	80.56%	70.84%	12.34%	4%	80.29%	77.76%	62.89%	97.22%	76.37%	2.4
SVM	96.11%	78.5%	48.23%	9.38%	1.5%	76.38%	73.7%	56.76%	92.26%	70.28%	1722
DNN	88.45%	91.24%	78.52%	46%	19%	85.33%	82.5%	78.01%	91.49%	84.21%	237
LR	92.81%	79.38%	64.93%	1.56%	3.0%	76.34%	73.58%	61.22%	89.23%	72.62%	63

**Table 7 sensors-22-00025-t007:** Detection performance of the proposed DSN.

Model	Normal	DoS	Probe	R2L	U2R	ACC	MACC	Recall	Precision	F1-Score	Time (s)
DSN	97.32%	90.7%	90.08%	49.02%	18%	90.41%	86.8%	78.82%	96.65%	86.83%	1652

**Table 8 sensors-22-00025-t008:** Comparison results based on NSL-KDD.

Author	Method	Data Set	ACC	MACC
Proposed method	DSN	NSL-KDD Test+	90.41%	86.8%
Pham [29]	Bagging	NSL-KDD Test+	/	84.25%
Kanakarajan [30]	GAR-forest	NSL-KDD Test+	/	85.05%
GAO [17]	Ensemble Voting	NSL-KDD Test+	/	85.2%
Tang [31]	SAAE-DNN	NSL-KDD Test+	87.74%	82.14%
Yang [14]	ICVAE-DNN	NSL-KDD Test+	85.97%	/
Proposed method	DSN	NSL-KDD Test-21	83.19%	79.2%
Yin [32]	RNN-IDS	NSL-KDD Test-21	/	64.67%
Yang [33]	MDPCA-DBN	NSL-KDD Test-21	/	66.18%
Tang [31]	SAAE-DNN	NSL-KDD Test-21	/	77.57%

## Data Availability

NSL-KDD Dataset. Available online: https://www.unb.ca/cic/datasets/nsl.html (accessed on 6 December 2021).

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
