# Peer review of "Deep Stacking Network for Intrusion Detection"

_sensors, 2021, doi:10.3390/s22010025_

Round 1
Reviewer 1 Report
- In this manuscript, the authors present an intrusion detection model based on deep stacking network. The authors combined the results of multiple classifiers to tackle the problem of unbalanced training data. Ensembles of multiple classifiers is a common way to deal with the problem of unbalanced training data. The authors should compare the results of the propose model with those of existing ensemble learning methods.
- Line 163, Page 5, The first paragraph in Section 3.3 is a repeat of the last paragraph in Section 3.2.
- Line 325, Page 10, “... structure with 4 hidden layers”, according to Figure 4, there are 5 hidden layers.
- Line 384, Page 12, “The six evaluation indicators ...”, however, only five indications are listed below.
- The title of this manuscript is “An Adaptive Deep Stacking Network for Intrusion Detection.” However, it’s not clear why the proposed method is “adaptive”.
Reviewer 2 Report
This paper is proposing the idea to detect an intrusion in the traffic data, therefore it can be used for an IDS system. The proposed algorithm consists of four algorithms decision tree, k-nearest neighbours, deep neural network, and random forests. It shows a better performance. However, it seems to be natural results as they use multiple classification methods which complement each other. Then, the accuracy can be improved.
This paper intensively compares the accuracy of detection methods. However, I could not see how the detection speed was. Because the proposed scheme applies multiple algorithms in the data set to improve accuracy, which may make detection slow. I think it is also important to show that the detection time is not significantly longer than the other. Also, as this paper proposes the scheme for the IDS system, the detection speed is also an important factor. It would be better to compare those in the paper.
The location of the tables and figures in the paper must be adjusted, properly.
Round 2
Reviewer 1 Report
As the authors have addressed all my comments, I suggest this manuscript to be accepted.
Reviewer 2 Report
Thank the authors for revising the paper in a short time. I believe that the requirements are properly reflected in the new version of the paper. It looks good to me, now.